# Prognostic Value and Clinical Significance of FGFR Genomic Alterations (GAs) in Metastatic Urothelial Cancer Patients

**DOI:** 10.3390/jcm11154483

**Published:** 2022-08-01

**Authors:** Elena Sevillano Fernández, Rodrigo Madurga de Lacalle, Juan Francisco Rodriguez Moreno, Arantzazu Barquín García, Mónica Yagüe Fernández, Paloma Navarro Alcaraz, María Barba Llacer, Miguel Quiralte Pulido, Jesús García-Donás Jiménez

**Affiliations:** 1HM CIOCC MADRID (Centro Integral Oncológico Clara Campal), Hospital Universitario HM Sanchinarro, HM Hospitales, 28050 Madrid, Spain; jfrodriguez@hmhospitales.com (J.F.R.M.); abarquin@hmhospitales.com (A.B.G.); moni_ca96@hotmail.com (M.Y.F.); pnavarro@hmhospitales.com (P.N.A.); mariabarba213@gmail.com (M.B.L.); miguel.quiralte@hotmail.com (M.Q.P.); jgarciadonas@hmhospitales.com (J.G.-D.J.); 2Departamento de Oncología Médica, Hospital Sanchinarro, Universidad San Pablo-CEU, CEU Universities, 28003 Madrid, Spain; 3Faculty of Experimental Sciences, Universidad Francisco de Vitoria, 28223 Madrid, Spain; rodrigo.madurga@ufv.es

**Keywords:** FGFR, independent factor, metastatic urothelial cancer

## Abstract

Fibroblast growth factor receptor (FGFR) genomic alterations (GAs) represent an actionable target, key to the pathogenesis of some urothelial cancers (UCs). Though FGFR GAs are common in noninvasive UC, little is known about their role in the metastatic(m) setting and response to therapy. This study aimed to assess the impact of FGFR alterations on sensitivity to systemic treatments and survival and to validate Bajorin’s and Bellmunt’s prognostic scores in mUC patients according to their FGFR status. We retrospectively analyzed data from 98 patients with tumor-sequenced UC who received treatment between January 2010 and December 2020. Up to 77 developed metastatic disease and were deemed the study population. Twenty-six showed FGFR GAs. A trend toward a better response to cisplatin and checkpoint inhibitors was suggested favoring FGFR GA tumors. FGFR GA patients who received an FGFR inhibitor as first-line had poorer responses compared with other options (20% vs. 68.4%, *p* = 0.0065). Median PFS was 6 vs. 5 months in the FGFR GA vs. FGFR WT cohort (*p* = 0.71). Median OS was significantly worse in the FGFR GA vs. FGFR WT cohort (16.2 vs. 31.9 months, *p* = 0.045). Multivariate analyses deemed FGFR GAs as a factor independently associated with the outcome (HR 2.59 (95% CI 1.21–5.55)). Bajorin’s model correctly predicted clinical outcomes in the whole study population but not in FGFR GA cases. FGFR GAs are a relevant biomarker in mUC that could condition the response to systemic therapy. New prognostic models, including this molecular determination, should be designed and validated.

## 1. Introduction

Bladder cancer is the eleventh most common cancer worldwide [1]. Genetic and environmental factors, such as smoking, exposure to occupational carcinogens, and infections, can lead to this disease [2,3]. The prognosis for patients with mUC remains poor. Around 25% of cases will present advanced disease at diagnosis or will develop metastases [1].

Bajorin’s prognostic model has been widely adopted in this disease and classifies mUC in three risk categories based on two prognostic factors: a Karnofsky performance status (KPS) of <80% and visceral metastases. Median survival varies between 9.3 and 33 months for poor and favorable risk patients, respectively [4].

Fortunately, significant improvements have been made in the therapeutic landscape of mUC. New options include checkpoint inhibitors (CPIs), FGFR inhibitors (FGFRis), and antigen-directed cytotoxic therapies (enfortumab vedotin or sacituzumab govitecan), which have revolutionized the management of this disease [5,6,7]. Thus, understanding the clinical implications of the molecular background of individual patients will help to define the optimal sequence of treatment options.

Fibroblast growth factor receptor gene alterations are common molecular findings in UC. Activating somatic mutations of FGFR3 have been detected in 50–70% of nonmuscle-invasive carcinomas and in 20% of mUC [8,9].

FGFR alterations have been identified as an early event in UC development.

Most of the missense mutations in the FGFR3 gene are clustered in three hotspots in exons 7, 10, and 15. The substitution of a serine for a cysteine at position 249 (S249C) is the most frequent pathogenic mutation. [10]. These alterations can lead to ligand-independent dimerization, autophosphorylation, and activation of the receptor [8].

Another frequent alteration is the fusion of FGFR3 to transforming acid coiled-coil containing protein 3 (TACC3), which leads to constitutive activation of the tyrosine kinase dominium [11]. FGFR3 amplification and alternative splicing are less frequent, and their functional implications are not yet fully understood [12].

Erdafitinib is the first FGFR antagonist approved by the FDA (2019) for the treatment of mUC with FGFR3 or FGFR2 mutations, but many others, such as rogaratinib, pemigatinib, and infigratinib, are being explored in different clinical settings [5,8,13].

Since FGFR GAs have become a cornerstone in mUC, we aimed to assess the value of such alterations and validate the current prognostic models in this particular population.

## 2. Materials (Patients) and Methods

### 2.1. Study Design and Patients

We designed a multicenter observational retrospective study. Eligible patients were adults diagnosed with UC from four hospitals in Madrid (Spain) between January 2010 and December 2020. The study protocol was approved by the institutional ethics committee, and patients provided written consent. Clinical records of patients who were deceased or lost to follow-up were reviewed following the instructions of the ethics committee. Only patients with tumor-sequenced metastatic disease who had received treatment were included in the analysis.

### 2.2. Genomic Analysis

Detection of FGFR mutations and fusions was performed on DNA isolated from formalin-fixed and paraffinized tumor tissue samples. The next-generation sequencing panel Foundation One^®^ or qualitative real-time polymerase chain reaction-based assays (*n* = 9) with TFGFR or QIAGEN therascreen^®^ were performed as routine practice.

### 2.3. Outcomes

The radiological response was based on the investigators’ judgment. PFS was calculated from the treatment start date to progression or death; OS was defined as the time from the treatment start date to the date of death from any cause. Patients without an event were censored at the date of last follow-up.

### 2.4. Statistical Analysis

Descriptive analysis was used to summarize the baseline characteristics of the study population and every cohort (FGFR GA or FGFR WT). Categorical variables were summarized as absolute frequency (%), while numerical variables were summarized as mean ± SD for normally distributed variables (Shapiro test, *p* > 0.05) or median (IQR) otherwise. Associations between FGFR alterations and clinical factors were analyzed using X^2^ or Fisher’s exact tests when needed for categorical variables and *t*-test or Mann–Whitney U test for numerical variables. Univariate Cox proportional hazard regression was performed to assess for correlations between FGFR altered genes, relevant baseline clinical and treatment characteristics, and survival. Cox proportional hazard models were applied for multivariate analysis. The hazard ratio (HR) and 95% confidence interval (CI) were calculated for every prognostic factor. Results were considered statistically significant at two-sided *p*-values of <0.05. PFS and OS were estimated using the Kaplan–Meier method, and statistical significance was assessed using the log-rank test. All analyses were performed using R (version 4.1.1).

## 3. Results

### 3.1. Overall Study Population

A total of 98 UC patients that underwent any form of FGFR testing were identified (Appendix A). Up to 77 developed metastatic disease and were deemed eligible for the study (Table 1; Figure 1A). The FGFR genomic alterations are summarized in Table 2.

### 3.2. Patient Characteristics

We compared baseline characteristics between FGFR GA and WT patients. Age, sex, ECOG, and smoking history did not differ significantly between FGFR status groups. Upper tract tumor location was more common in the FGFR GA cohort (*p* = 0.037), and more nephroureterectomies were performed consequently (*p* = 0.0076). FGFR GAs were associated with more pT4 stages (*p* = 0.022). (Table 1).

### 3.3. First-Line Treatment Outcome

First-line systemic treatment consisted of cisplatinum-based chemotherapy (*n* = 32), checkpoint inhibitors (CPIs) (*n* = 21), FGFRis (*n* = 5), and others (*n* = 19) in 26 FGFR GA and 51 FGFR WT patients.

In the whole study population, the ORR was 50%, including 7.1% (*n* = 5) CRs and 43% (*n* = 30) PRs. Eight additional patients (11.4%) achieved stable disease (SD) as the best response, with a disease control rate (DCR) of 61.5%.

The ORR to first-line cisplatin-based therapy was 59.4% and 38.1% with CPI. FGFR GA patients showed a trend toward a better response to cisplatin and checkpoint inhibitors with regards to the WT cohort. No statistically significant differences in PFS and OS according to FGFR status were found in the first-line therapy subgroups (Appendix A).

Five FGFR GA patients who received a first-line FGFRi had poorer responses compared with other first-line options (20% vs. 68.4%, *p* = 0.0065). (Table 3).

### 3.4. Progression-Free Survival and Overall Survival

PFS and OS information was available in 76 of the 77 patients.

Median PFS after first-line treatment for the whole study population was 6 months in the FGFR mut/fus cohort and 5 months in the FGFR WT patient group (*p* = 0.71) (Figure 1B).

Median OS for the whole study population was 20 months and was significantly worse in FGFR GA (mutation/fusion) vs. FGFR WT tumors (16.2 vs. 31.9 months, *p* = 0.045; Figure 1C). When stratified by metastases location (liver, bone, visceral, and lymph node), there were no significant differences between FGFR GA and WT patients (Figure 1D).

No significant differences were found in OS in patients with variant histologies (*n* = 7) (*p* = 0.67) or when comparing urothelial with squamous (*n* = 5, *p* = 0.94). There were also no differences in clinical outcomes among patients with bladder and UTUC (*p* = 0.88) or when stratifying by FGFR status (Appendix A).

### 3.5. Clinical and Molecular Prognostic Factors: Univariate and Multivariate Survival Analysis

The log-rank test revealed that FGFR GA tumors were associated with a shorter OS, and the univariate Cox regression model confirmed this result (HR 1.87 (95%CI 1.01 to 3.48)). Univariate analyses revealed other variables associated with survival: the presence of visceral metastases (HR 4.87 (95% CI 1.48 to 16.0)) and ECOG > 1 (HR 2.79 (95% CI 1.29 to 6.00)). A multivariable model, including the abovementioned variables as well as age, tumor location, and treatment, showed that FGFR GA was independently associated with survival (HR 2.59 (95% CI 1.21 to 5.55)). In addition, age (HR 1.03 (95% CI 1.00 to 1.07)), visceral metastases (HR 11.4 (95% CI 2.56 to 50.9)), and ECOG >1 (HR 6.4 (95% CI 2.43 to 16.9)) were independently associated with survival (Table 4).

We aimed to analyze the validity of classical prognostic models in the study population, subdivided according to FGFR genomic alterations. First, we explored Bajorin’s model that considers visceral metastases and KPS < 80% as poor risk factors. Median OS in patients treated with first-line platinum-based chemotherapy with zero, one, or two risk factors were: not reached, 22.9, and 28 months, respectively (*p* = 0.14). The same analysis was performed in patients treated with first-line CPI with mOS: not reached, 25.6, and <1 month, respectively (*p* = 0.0001). When we analyzed Bajorin’s criteria according to FGFR status, they were fulfilled in FGFR WT patients (*p* = 0.00058) but did not reach statistical significance in FGFR GA patients (*p* = 0.11) (Figure 1E).

Subsequently, we tested Bellmunt’s model in the second line, which also includes anemia. We observed a trend toward better survival in chemotherapy-treated patients with no risk factors (*p* = 0.085). No trend was found in patients who received CPI (*p* = 0.99) or in FGFR GA patients (*p* = 0.3) (Figure 1F).

### 3.6. Interaction of FGFR GAs with Additional Biomarkers

In order to further characterize the associations among FGFR GAs with the outcome and other molecular biomarkers, we added four FGFR amplified cases to the FGFR mut/fus cohort (*n* = 30).

We explored the relationship between the FGFR genomic alteration type and treatment response (cisplatin, CPI, FGFRi). No significant differences in response rates were observed according to the aberration subtype (Appendix A). Regarding PFS and OS, the outcomes were equally poor irrespective of the genomic alteration (Figure 2A).

Subsequently, we analyzed the interactions between FGFR GAs, the potential molecular biomarkers of the response to CPI (PD-L1, TMB ), and other mutations of interest, including CDKN2A/B loss and TERT promoter mutations. Phi correlation matrixes were calculated, and a Venn diagram was constructed for 30 FGFR GA tumors to determine the association with clinical outcomes (Figure 2C,D).

The prevalence, by NGS, of TERT promoter mutations was 11/30 (36%), and 9/30 (30%) had a loss of CDKN2A/B. PD-L1 positive immunohistochemistry expression was present in 8/30 (26%), and TMB results were available in 11/30 (36%) of the samples.

FGFR GA PD-L1 CPS ≥ 1 tumors were associated with worse clinical outcomes (*p* = 0.028). We also observed a trend toward a worse OS in FGFR GA patients with associated TERT promoter mutations or a TMB >10 (Figure 2B).

CDKN2A/B loss and TERT promoter mutations were positively correlated (Phi 0.56), as was PD-L1 (CPS ≥ 1) and a TMB > 10 (Phi 0.31). Another significant association was found between FGFR mutations and TERT promoter mutations with a medium–strong value (Phi 0.34); the correlation was weaker for FGFR mutations and CDKN2A/B loss (Phi 0.24). Interestingly, these relationships were not observed in the FGFR-translocated or amplified cases. A weak association between PD-L1 CPS ≥ 1 and FGFR mutated (Phi 0.03) or amplified (Phi 0.15) genes was also observed (Figure 2C,D).

## 4. Discussion

Growing evidence suggests that FGFR GAs are associated with worse outcomes in different tumor types [14,15,16]. In spite of the general correlation of FGFR GA with lower grades and stages of nonmuscle-invasive UC, there is no evidence to support that FGFR GAs are associated with a favorable phenotype once urothelial carcinoma advances [17].

Thus, we aimed to assess the value of FGFR GAs in mUC, as it remains unknown [18]. Among the 77 mUC patients analyzed, 26 presented FGFR GAs, and 51 were FGFR WT. Patients showing FGFR GAs had a significantly worse overall survival compared with the WT cohort.

In the multivariate analysis, an ECOG PS > 1, visceral metastasis, and treatment regimens other than cisplatin-based were confirmed as independent prognostic factors. Moreover, FGFR GAs were significantly associated with worse OS (HR 2.59 (95% CI 1.21–5.55)).

Unfortunately, since all patients had received several types of treatment, this study could not differentiate between a prognostic or predictive role of FGFR GAs. Thus, the outcome could be influenced by the response to treatments rather than by the biological action of these alterations.

The impact of FGFR GAs was irrespective of the type of molecular alteration (mutation, fusion, amplification), supporting the driver role of this gene.

Studies are controversial regarding FGFR GA and CPI responsiveness [19,20,21]. Though our results did not show a statistically significant difference between FGFR GA and WT tumors, a trend toward a better response to both treatments was suggested favoring GA tumors. A statistically significant increase in PFS after first-line treatment in the FGFR GA cohort was also observed. This is in contrast with the worse OS for this population and with the worse OS for this population. Therefore, FGFR GA UC might represent a distinct biological entity that correlates with an aggressive phenotype once the UC becomes advanced.

A rapid progression after perioperative treatment could explain these differences, which reinforces the aggressiveness of its nature. These results are in line with prior communications, including muscle-invasive bladder cancer, in which alterations in the FGFR gene have been associated with inferior responses to neoadjuvant platinum-based chemotherapy and a higher recurrence rate [22]. Intriguingly, this difference did not lead to a worse OS as in our study, which included only metastatic UC.

Hence, an adequate sequenced approach must be considered in this population, and a phase III clinical trial is underway (NCT03390504).

Interestingly, first-line treatment with an FGFRi showed worse response rates compared with other therapeutic approaches in FGFR GA patients. However, only five cases presented this condition.

Surprisingly, we did not find significant differences in the ORR to any of the treatments according to the FGFR genomic alteration type, as different susceptibilities to erdafitinib have been described in FGFR3 mutations (ORR: 49%), fusions (ORR: 16%), and for FGFR3:TACC3v1 (ORR: 36%) [5].

Since the sample size of our study was small, larger studies should elucidate the specific role of the different genomics alterations described in FGFR in mUC.

Clinical prognostic scores are commonly used in mUC to stratify patients and predict outcomes. Bajorin showed that a poor KPS and visceral metastases were independently associated with worse outcomes [4]. Bellmunt reported that an ECOG PS ≥ 1, anemia, and liver metastasis predict OS in platinum-refractory patients [23].

Both models have been widely adopted. However, the treatment landscape of mUC has significantly changed, and, currently, FGFR molecular alterations determine the choice of the treatment strategy.

In this study, Bajorin’s model correctly predicted clinical outcomes in the whole study population, and its accuracy even improved when restricted to the FGFR WT cases.

These results lead to the notion that classical algorithms could work for FGFR WT cases but not for FGFR GAs tumors. However, such findings should be confirmed in larger studies since none of the FGFG GA patients had ≥2 poor prognostic factors in this study, likely due to the low numbers.

Although UC has been considered a single entity, the evidence points toward distinct biological differences between UTUC and UBC. [24,25]. Nevertheless, the outcomes were equivalent irrespective of the tumor location, and the multivariate analysis was not significant. Outcomes in UTUC tumors harboring FGFR GA vs. FGFR WT were also similar.

Finally, we explored the incidence and potential role of additional biomarkers and genomic alterations in the FGFR GA population.

FGFR mutations have been associated with a lower PD-L1 expression, decreased T-cell infiltration, and a predominantly luminal-papillary subtype [26,27,28,29]. In total, 26% of the FGFR GA patients had a PD-L1 CPS ≥ 1 tumoral expression and were associated with a worse outcome. This finding supports the importance of several ongoing clinical trials combining FGFRi and CPI that could specifically benefit these poor-prognosis patients [30].

A high TMB is commonly used as a predictive biomarker for CPI [31]. Unfortunately, prior authors have communicated a low TMB in the FGFR GA tumors [32]. In our study, 4/10 FGFR GA patients had >10 mutations/Mb. However, they did not present a different clinical course, and conclusions must be taken with caution due to the limited sample size.

TERT promoter mutations are frequent GAs in UC and indicate poor prognostic and increased recurrence rates [33,34]. Interestingly, we identified a correlation between FGFR GAs and TERT promoter mutations, although no impact on response to therapy was observed. TERT mutations also seem to correlate with a higher TMB and PD-L1 expression, which may predict immunotherapy response in the FGFR GA population. [35].

CDKN2A is a tumor suppressor that renders the retinoblastoma inactive [36]. We observed a positive association between FGFR mutations and CDKN2A/B loss, as described with the TCGA [36,37]. Such alterations showed a trend toward increased survival. CDKN2A loss may cause resistance to CPI in UC [38], as CDKN2A and CD274 (encoding PD-L1) are both encoded in p9 chromosome 9 [39]. However, further validation is required.

Finally, it must be highlighted that some pathological and molecular factors previously associated with the outcome are missing in our study. As an example, FGFR GAs are overrepresented in the luminal subtype, which seems to have a better prognosis [28,29]. In addition, lymphovascular invasion and fibronectin expression could play a role when determining the best therapeutic option for every patient [40,41].

## 5. Conclusions

Our hypothesis-generating real-world data analysis suggests that the aberrations in FGFR may be an independent biomarker in mUC that should be included in new prognostic models.

## Figures and Tables

**Figure 1 jcm-11-04483-f001:**
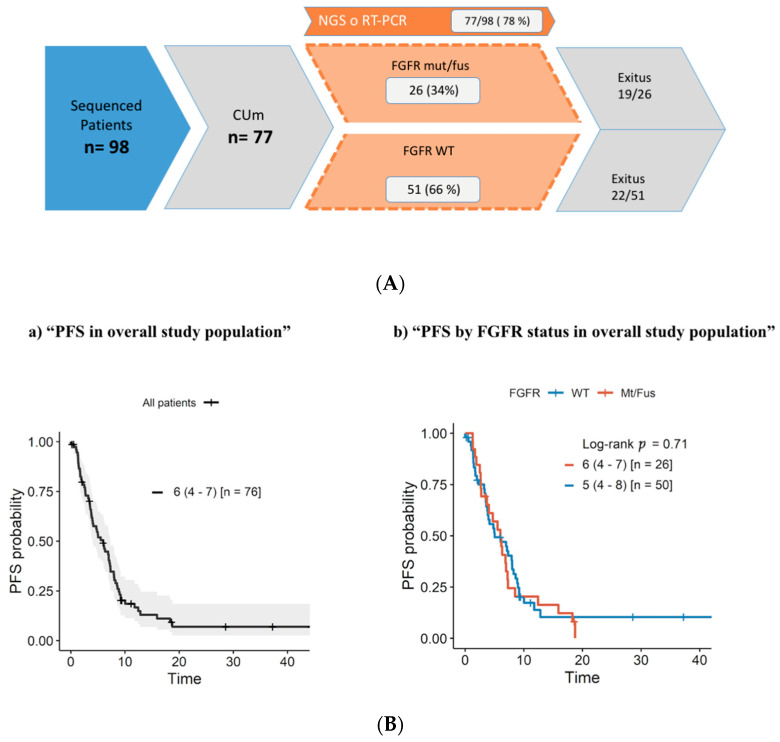
(**A**) Study population flowchart. (**B**) Progression-free survival according to FGFR genomic alterations. (**C**) Overall survival according to FGFR status. (**D**) Overall survival according to FGFR status in patients with liver metastases, bone metastases, visceral metastases, or lymph node only metastases. (**E**) Bajorin’s criteria in patients treated with first-line cisplatin-based therapy (**a**) and immune therapy (**b**); Bajorin’s criteria in FGFR WT patients (**c**); Bajorin’s criteria in FGFR mut/fus patients (**d**). (**F**) Bellmunt’s criteria for overall survival in patients treated with second-line chemotherapy (**a**) and immune therapy (**b**); Bellmunt’s prognostic factors in FGFR WT patients (**c**) and FGFR GA patients (**d**).

**Figure 2 jcm-11-04483-f002:**
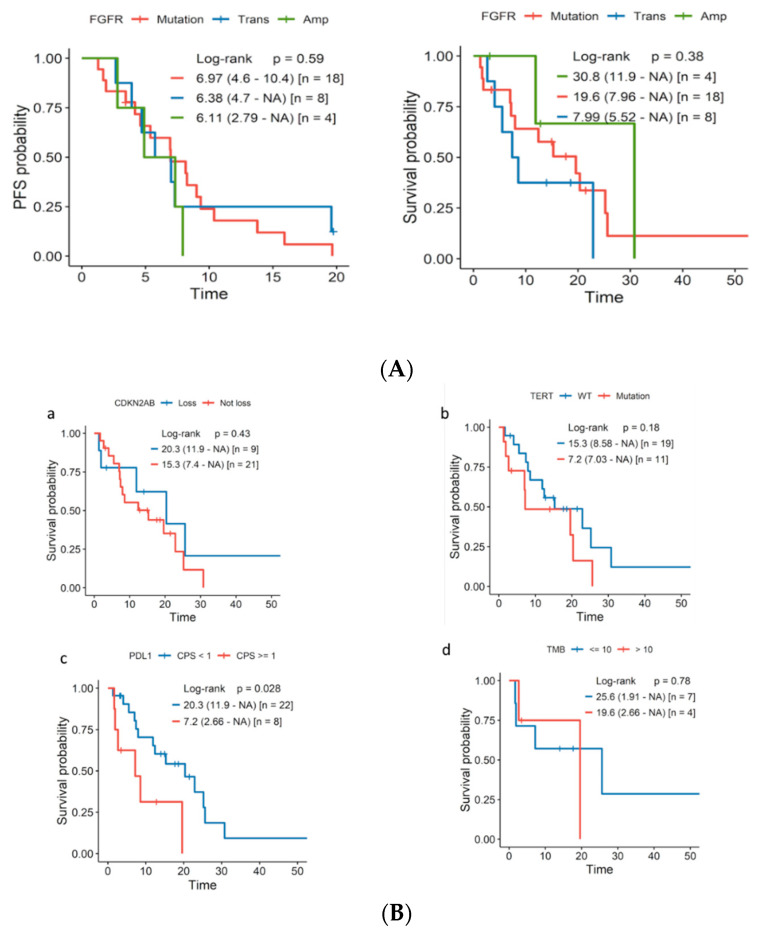
(**A**) PFS or OS according to the type of FGFR genomic alteration: translocation/fusion ( Trans) and amplification (Amp) cases were added. (**B**) Overall survival according to additional biomarkers: (**a**) CDKN2A/B loss; (**b**) TERT promoter mutations; (**c**) PD-L1 (CPS < 1 vs. CPS ≥ 1); (**d**) TMB (≤10 vs. >10). (**C**) Phi coefficient assessing the correlation between different biomarkers in the FGFR GA population. Phi correlation. Heat map: the phi index ranges from −1 to +1; red indicates a positive correlation (darker red indicates a stronger correlation between two biomarkers); blue indicates a negative correlation; white (phi = 0) represents no correlation. (**D**) Venn diagram representing the superposition of the expression of different biomarkers: expression of PD-L1 ≥ 1, FRGR genomic alterations, loss of cyclin-dependent kinase inhibitor (CDKN2A/B), and TERT promoter mutation.

**Table 1 jcm-11-04483-t001:** Study population demographics (*n* = 77) and comparison between FGFR GA and WT patients.

Variable	Modality	Metastatic	Mt/Fus (*n* = 26)	WT (*n* = 51)	*p*-Value
**Age**	Median (IQR)	69 (62–76)	69 (63–77)	69 (61–75)	0.45
**Sex**	Male	55 (71.4%)	17 (65.4%)	38 (74.5%)	0.57
	Female	22 (28.6%)	9 (34.6%)	13 (25.5%)	
**ECOG PS**	0	26 (33.8%)	9 (34.6%)	17 (33.3%)	0.15
	1	25 (32.4%)	14 (53.8%)	11 (21.6%)	
	2	2 (2.6%)	0 (0%)	2 (3.9%)	
	Not available	24 (31.2%)	3 (11.5%)	21 (41.2%)	
**Smoking**	Never smoked	16 (25.4%)	7 (26.9%)	9 (17.6%)	0.55
	Current smoker	9 (14.3%)	3 (11.5%)	6 (11.8%)	
	Former smoker	38 (60.3%)	11 (42.3%)	27 (52.9%)	
	Not available	14 (18.2%)	5 (19.2%)	9 (17.6%)	
**Tumor location**	Bladder	62 (80.5%)	17 (65.4%)	45 (88.2%)	0.037
	Nonbladder	15 (19.5%)	9 (34.6%)	6 (11.8%)	
**Surgery**	No	4 (5.2%)	1 (3.8%)	3 (5.9%)	1
	Yes	73 (94.8%)	25 (96.2%)	48 (94.1%)	
**Surgery extension**	Cystectomy (RC)	38 (49.3%)	7 (28.0%)	31 (64.6%)	0.0076
	Nephroureterectomy/nephrectomy(NU)	12 (15.6%)	8 (32.0%)	4 (8.3%)	
	NU + RC	3 (3.9%)	1 (4.0%)	2 (4.2%)	
	TURBT	20 (26%)	9 (36.0%)	11 (22.9%)	
**Lymphadenectomy**	No	36 (46.8%)	15 (57.7%)	21 (41.2%)	0.37
	Yes	38 (49.3%)	11 (42.3%)	27 (52.9%)	
	Not available	3 (3.9%)	0 (0.0%)	3 (5.9%)	
**Bladder preservation**	Radiotherapy	4 (5.2%)	2 (7.7%)	2 (3.9%)	0.86
	Chemo-radiotherapy	6 (7.8%)	2 (7.7%)	4 (7.8%)	
	No	67 (87.0%)	22 (84.6%)	45 (88.2%)	
**pT**	1	7 (9.1%)	3 (11.5%)	4 (7.8%)	0.022
	2	28 (36.4%)	8 (30.8%)	20 (39.2%)	
	3	28 (36.4%)	6 (23.1%)	22 (43.1%)	
	4	13 (16.8%)	9 (34.6%)	4 (7.8%)	
	Not available	1 (1.3%)	0 (0.0%)	1 (2.0%)	
**pN**	0	14 (36.8%)	3 (27.3%)	11 (40.7%)	0.81
	1	11 (28.9%)	4 (36.4%)	7 (25.9%)	
	2	12 (31.6%)	4 (36.4%)	8 (29.6%)	
	3	1 (2.6%)	0 (0%)	1 (3.7%)	
**Grade**	2	2 (2.6%)	1 (3.8%)	1 (2%)	1
	3	73 (94.8%)	25 (96.2%)	48 (94.1%)	
	NA, n (%)	2 (2.6%)	0 (0.0%)	2 (3.9%)	
**Variant histologies**	Transitional cells	70 (90.9%)	23 (88.5%)	47 (92.2%)	0.71
	Squamous	5 (6.5%)	2 (7.7%)	3 (5.9%)	
	Anaplastic	1 (1.3%)	1 (3.8%)	0 (0%)	
	Neuroendocrine	1 (1.3%)	0 (0%)	1 (2%)	
**Perioperative chemotherapy**	No	52 (67.5%)	16 (61.5%)	36 (70.6%)	0.67
	Neoadjuvant	12 (15.6%)	5 (19.2%)	7 (13.7%)	
	Adjuvant	13 (16.9%)	5 (19.2%)	8 (15.7%)	
**Liver metastases**	No	61 (79.2%)	22 (84.6%)	39 (76.5%)	0.7
	Yes	15 (19.5%)	4 (15.4%)	11 (21.6%)	
	Not available	1 (1.3%)	0 (0.0%)	1 (2.0%)	
**Bone metastases**	No	46 (59.7%)	15 (57.7%)	31 (60.8%)	0.91
	Yes	30 (39.0%)	11 (42.3%)	19 (37.3%)	
	Not available	1 (1.3%)	0 (0.0%)	1 (2.0%)	
**Lymph node only metastases**	Yes	15 (19.5%)	5 (19.2%)	10 (19.6%)	1
**Visceral metastases**	Yes	60 (77.9%)	21 (80.8%)	39 (76.5%)	1
	Not available	1 (1.3%)	0 (0.0%)	1 (2.0%)	
**First-line treatment for mUC**	Cisplatin-based	33 (42.8%)	11 (42.3%)	22 (44.0%)	
	Checkpoint inhibitors	23 (29.9%)	5 (19.2%)	18 (36.0%)	
	Carboplatin-based	8 (10.4%)	2 (7.7%)	6 (12.0%)	
	FGFR inhibitor	5 (6.5%)	5 (19.2%)	0 (0.0%)	
	Vinflunine	3 (3.9%)	1 (3.8%)	2 (4.0%)	
	Best supportive care	2 (2.6%)	2 (7.7%)	0 (0.0%)	
	Paclitaxel	1 (1.3%)	0 (0.0%)	1 (2.0%)	
	Surgery	1 (1.3%)	0 (0.0%)	1 (2.0%)	
	Not available	1 (1.3%)	(0%)	1 (2.0%)	

**Table 2 jcm-11-04483-t002:** FGFR genomic alterations.

TYPE OF FGFR	N (26)	%
GENOMIC ALTERATION (Number of Cases)		
**MUTATION**	15	57.7%
-FGFR3 S249C (13)		
-FGFR3 S249C // S783 frameshift mutation (1)		
-FGFR3 S249C // H349D (1)		
**FUSION**	6	23.1%
-FGFR1-FGFR1 (1)		
-FGFR3-TACC3 (3)		
-FGFR2-OFD1 (1) -FGFR2-AFF3 (1)		
**MUTATION + AMPLIFICATION**	2	7.7%
-FGFR3 S249C + FGFR1 amplification (1)		
-FGFR3 S249C + FGFR1 amplification (1)		
**MUTATION + FUSION**	1	3.8%
-FGFR3 R248C // S249C + FGFR3-TACC3 (1)		
**FUSION + AMPLIFICATION**	2	7.7%
-FGFR2-RTKN2 + FGFR2 amplification (1) -FGFR3-TACC3 + FGFR1 amplification (1)		

Next-generation sequencing with the Foundation One^®^ test was performed in 68 cases, and qualitative real-time polymerase chain reaction-based assay TFGFR or QIAGEN therascreen^®^ tests in 9. TFGFR or QIAGEN therascreen^®^ tests evaluated somatic mutations within the FGFR3 gene: R248C, S249C, G370C, and Y373C, and fusions: FGFR3-TACC3v3, FGFR3-TACC3v1, FGFR3-BAIAP2L1, FGFR2-BICC1, and FGFR2-CASP7.

**Table 3 jcm-11-04483-t003:** Treatment response to first-line therapy according to FGFR status and specific therapy according to the RECIST criteria v1.1.

Population	Treatment (n)	Type of Response						
		CR	PR	SD	PD	ORR	*p*-Value *	*p*-Value ORR
**Overall**	**Any (70)**	5 (7.1%)	30 (42.9%)	8 (11.4%)	27 (38.6%)	35 (50.0%)		
**FGFR WT**		3 (6.5%)	18 (39.1%)	5 (10.9%)	20 (43.5%)	21 (45.7%)	0.71	0.57
**FGFR mut/fus**		2 (8.3%)	12 (50.0%)	3 (12.5%)	7 (29.2%)	14 (58.3%)		
**Overall**	**Cisplatinum-based (32)**	2 (6.3%)	17 (53,1%)	5 (15.6%)	8 (25.0%)	19 (59.4%)		
**FGFR WT**		1 (4.8%)	10 (47.6%)	3 (14.3%)	7 (33.3%)	11 (52.4%)	0.43	0.45
**FGFR mut/fus**		1 (9.1%)	7 (63.6%)	2 (18.2%)	1 (9.1%)	8 (72.7%)		
**Overall**	**Immunotherapy (21)**	2 (9.5%)	6 (28.6%)	2 (9.5%)	11 (52.4%)	8 (38.1%)		
**FGFR WT**		1 (6.3%)	4 (25.0%)	2 (12.5%)	9 (56.3%)	5 (31.3%)	0.59	0.33
**FGFR mut/fus**		1 (20.0%)	2 (40.0%)	0 (0.0%)	2 (40.0%)	3 (60.0%)		
**FGFR mut/fus**	**FGFR inhibitors (5)** **Other (19)**	0 (0.0%)	1 (20.0%)	0 (0.0%)	4 (80.0%)	1 (20.0%)	0.065	0.12
		2 (10.5%)	11 (57.9%)	3 (15.8%)	3 (15.8%)	13 (68.4%)		

* (ref) *p*-values for the four types of response and for the proportion of ORR were obtained using Chi-squared tests or Fisher exact test when necessary.

**Table 4 jcm-11-04483-t004:** Univariate and multivariate analysis for prognostic factors and overall survival.

		Univariate		Multivariate	
Variable	Modality	HR	95% CI	HR	95% CI
**Age**	(continuous)	1.02	0.992–1.05	1.03	1.00–1.07
**Location**	Nonbladder	1 (ref) *	-	1 (ref)	-
	Bladder	1.07	0.47–2.42	1.39	0.56–3.48
**Treatment**	Cisplatin	1 (ref)	-	1 (ref)	-
	Immunotherapy	1.32	0.60–2.90	2.40	0.97–5.90
	Other	1.71	0.84–3.48	3.17	1.38–7.24
**Visceral metastases**	No	1 (ref)	-	1 (ref)	-
	Yes	4.87	1.48–16.0	11.4	2.56–50.9
**ECOG > 1**	No	1 (ref)	-	1 (ref)	-
	Yes	2.79	1.29–6.00	6.40	2.43–16.9
**FGFR**	Wild-type	1 (ref)	-	1 (ref)	-
	Mutated	1.87	1.01–3.48	2.59	1.21–5.55

* (ref) denotes the category used as reference.

## Data Availability

The data presented in this study is available in this article.

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
