# Peer review of "Prognostic Value and Clinical Significance of FGFR Genomic Alterations (GAs) in Metastatic Urothelial Cancer Patients"

_jcm, 2022, doi:10.3390/jcm11154483_

Round 1
Reviewer 1 Report
The authors investigated the prognostic role of FGFR alterations in metastatic urothelial carcinoma patients. A further molecular differentiation of those patients is important for an individual decision making and FGFR(3) is of high relevance in this tumor disease. However, the study is lacking some important analyses and more detailed discussion:
The ORR to cisplatin-based chemotherapy and ICI seems to be higher in the FGFR mut/fus group, although it is not significant. At least, this should be mentioned. This is, on one hand, in contrast to published data and also to the Km-curves concerning survival, on the other hand.
K-M curves are used to estimate prognosis. However, all patients are treated, and by different therapies. Therefore, prognostic evaluation is impossible in this population. General prognostic factors can be evaluated only in untreated populations. In this population it is impossible to distinguish between prognostic and predictive factors. PFS and OS concerning FGFR should be analyzed in therapy subgroups.
It is surprisingly that in the FGFR group no Bajorin 3 patients can be found. Is that due to the low patient number?
The discussion should be more focused on critical analysis concerning the prognostic value. It cannot be concluded from the presented data that FGR is a prognostic marker as discussed above. The poor prognosis might be based on poor response to treatment. The authors should also discuss molecular subtypes. In the new consensus classification, the subtype (luminal papillary) with FGFR3 mutations has the best prognosis. The better prognosis is also shown in other studies ( van Rhin, Teo). It has to be mentioned that these studies have been done in MIBC and not exclusively in mUC. However, they also conclude that the outcome is influenced by a much lower (or no?) response to (peri)operative chemotherapy. In addition, cases with mutations should be analyzed separately as it was shown that other alterations, especially higher expression, is not relevant. Taking together, from the presented data it cannot be concluded that FGFR alterations are poor prognostic markers.
Author Response
Please see the attachment
To the editors:
Thanks for considering our paper for publication. We have read with great interest the comments by both reviewers and think that will greatly help to improve our paper.
Please, see below our point by point response
1-The ORR to cisplatin-based chemotherapy and ICI seems to be higher in the FGFR mut/fus group, although it is not significant. At least, this should be mentioned. This is, on one hand, in contrast to published data and also to the Km-curves concerning survival, on the other hand.
We agree and have modified the discussion accordingly:
Line 254, that earlier read: “Our findings suggest that FGFR GA and FGFR WT tumors respond similarly to CPI and cisplatin-based treatments” now reads: Though our results did not show a statistically significant difference between FGFR GA and WT tumors, a trend towards a better response to both treatments was suggested favoring GA tumors…This is in contrast with the worse OS for this population.”
Additionally, we have moved the paragraph that mentioned the conflicting results reported in literature (that reads: “A rapid progression after…”) to line 261
Finally, we have add, in line 266 the phrase: “Intriguingly, this difference did not lead to a worse OS“ pointing to the difference with our results
2-K-M curves are used to estimate prognosis. However, all patients are treated, and by different therapies. Therefore, prognostic evaluation is impossible in this population. General prognostic factors can be evaluated only in untreated populations. In this population it is impossible to distinguish between prognostic and predictive factors.
We agree that it is not possible to distinguish between a prognostic or predictive effect in our study. However, it must be highlighted that similar designs like Bajorin JCO 1999 and Bellmunt JCO 2020 also used the term “prognostic factor” in similar circumstances.
In order to state clearly the limitations of our study, as suggested by reviewer 1, we have made relevant modifications:
The term “prognostic” has been replaced in the following lines:
- Line 37 earlier read: FGFR GAs were found to be an independent prognostic factor (HR 2.59 [95% CI 1.21-5.55]). Now reads: “FGFR GAs as a factor independently associated with outcome (HR 2.59 [95% CI 1.21-5.55]).
- Line 45 earlier read: KEYWORDS: FGFR as a prognostic factor in mUC”. Now reads: “KEYWORDS: FGFR as an independent factor associated with outcome in mUC”
- Line 92 earlier read: Since FGFR GAs have become a cornerstone in mUC, we aimed to assess the prognostic value of such alterations and to validate the current prognostic models in this particular population. Now reads: “Since FGFR GAs have become a cornerstone in mUC, we aimed to assess the value of such alterations and to validate the current prognostic models in this particular population. “
- Line 239 earlier read: “Thus, we aimed to assess the prognostic value of FGFR GAs in mUC, as…” Now reads: “Thus, we aimed to assess the value of FGFR GAs in mUC, as…”
- Line 251 earlier read: “The prognostic impact of FGFR GAs was irrespective of the type of molecular alteration…” Now reads: “The impact of FGFR GAs was irrespective of the type of molecular alteration…”
- Line 331 earlier read:” FGFR may be an independent prognostic biomarker….” Now reads: “FGFR may be an independent biomarker”
3- PFS and OS concerning FGFR should be analyzed in therapy subgroups.
We agree that every information regarding treatment utility in tumors with FGFR GAs is interesting and relevant for readers. Thus, we have added such analysis as suggested. Due to the low numbers, they are represented as supplemental material (Supp Figures 4 A-D) but could be moved to the main text if agreed by the editors.
4-It is surprisingly that in the FGFR group no Bajorin 3 patients can be found. Is that due to the low patient number?
We agree and have modified the discussion accordingly. Line 290 earlier read: “However, such findings should be confirmed in larger studies, since none of the FGFG GA patients had ≥ 2 poor prognostic factors in this study” and now reads: “However, such findings should be confirmed in larger studies, since none of the FGFG GA patients had ≥ 2 poor prognostic factors in this study, likely due to the low numbers.”
5-The discussion should be more focused on critical analysis concerning the prognostic value. It cannot be concluded from the presented data that FGR is a prognostic marker as discussed above. The poor prognosis might be based on poor response to treatment.
We agree on the importance of adequately discussing the relevance of FGFR GAs since they have led to the first targeted therapy in urothelial cancer. Thus, we have added all the recommendations to the discussion:
- Line 241 earlier read “Patients showing FGFR GAs had a significantly worse outcome compared to the WT cohort” Now reads: Patients showing FGFR GAs had a significantly worse overall survival compared to the WT cohort
As stated above we have modified the text since the prognostic value of FGFR cannot be demonstrated in our study (see response to point #1)
Additionally, we have included the phrase Line 247: “Unfortunately, since all patients had received several types of treatment, this study could not differentiate between a prognostic or predictive role of FGFR GAs. Thus, outcome could be influenced by response to treatments rather than by the biological action of these alterations.”
6-The authors should also discuss molecular subtypes. In the new consensus classification, the subtype (luminal papillary) with FGFR3 mutations has the best prognosis.
We agree and have included the phrase line 324 “Finally, it must be highlighted that some pathological and molecular factors, previously associated with outcome, are missing in our study. As an example, FGFR GAs is overrepresented in the luminal-papillary subtype, that seems to have a better prognosis.”
7-The better prognosis is also shown in other studies ( van Rhin, Teo). It has to be mentioned that these studies have been done in MIBC and not exclusively in mUC. However, they also conclude that the outcome is influenced by a much lower (or no?) response to (peri)operative chemotherapy.
We agree and have specifically included Teo et al as reference #22. Also line 262 that earlier read: These results are in line with prior communications where alterations in FGFR gene have been associated with inferior responses to neoadjuvant platinum-based chemotherapy and a higher recurrence rate” now reads: “These results are in line with prior communications, also including muscle invasive bladder cancer, where alterations in FGFR gene have been associated with inferior responses to neoadjuvant platinum-based chemotherapy and a higher recurrence rate”
8-In addition, cases with mutations should be analyzed separately as it was shown that other alterations, especially higher expression, is not relevant.
We agree and have added the phrase line 276: Since numbers in our study were small, larger series should elucidate the specific role of the different genomics alterations described in FGFR in mUC.
9-Taking together, from the presented data it cannot be concluded that FGFR alterations are poor prognostic marker
We agree and, as mentioned, we have modified text accordingly

Reviewer 2 Report
Dear Authors,
i reviewed with interest the paper entitled "Prognostic Value and Clinical Significance of FGFR Genomic 2 Alterations (GA) in Metastatic Urothelial Cancer Patients".
I found the article quite interesting, with good methods and well presented.
I have few comments to report:
1. Please discuss how the improved ORR and PFS in the mutated cohort matched with the worse OS in the same cohort as compared to the counterpart.
2. The abstract should be improved. Results are not clearly described
3. In the MVA the outcomes analyzed is not clear.
4. Please consider expanding the reference list including the following valuable papers: doi: 10.1016/j.urolonc.2022.01.014 ; doi: 10.1177/1756287221995683 ;
Author Response
Please see the attachement
To the editors:
Thanks for considering our paper for publication. We have read with great interest the comments by both reviewers and think that will greatly help to improve our paper.
Please, see below our point by point response
- Please discuss how the improved ORR and PFS in the mutated cohort matched with the worse OS in the same cohort as compared to the counterpart.
We agree and have modified the discussion accordingly:
Line 254, that earlier read: “Our findings suggest that FGFR GA and FGFR WT tumors respond similarly to CPI and cisplatin-based treatments” now reads: Though our results did not show a statistically significant difference between FGFR GA and WT tumors, a trend towards a better response to both treatments was suggested favoring GA tumors…This is in contrast with the worse OS for this population.”
Additionally, we have moved the paragraph that mentioned the conflicting results reported in literature (that reads: “A rapid progression after…”) to line 261
Finally, we have add, in line 266 the phrase: “Intriguingly, this difference did not lead to a worse OS“ pointing to the difference with our results
- The abstract should be improved. Results are not clearly described
We have rewritten the abstract as suggested by reviewer # 1 and # 2 (lines 17 to 43).
- In the MVA the outcomes analyzed is not clear.
We thank reviewer #2 to point out this issue. We have rewritten paragraph that describes MVA results to make it clearer (lines 178 to 187).
- Please consider expanding the reference list including the following valuable papers: doi: 10.1016/j.urolonc.2022.01.014 ; doi: 10.1177/1756287221995683
We agree the interest of such references and have added both in line 327: Also linfovascular invasion and fibronectin expression could play a role when determining the best therapeutic option for every patient.

Round 2
Reviewer 1 Report
The authors replied in an adequate way to the comments and critical points. There is one point that have to be clarified.
7.: These results are in line with prior communications, also including muscle invasive bladder cancer…
You should mention that you included only metastatic UC.
Author Response
"Please see the attachment"
7.: These results are in line with prior communications, also including muscle invasive bladder cancer…You should mention that you included only metastatic UC.
We agree and have modified the sentence accordingly:
Line 266, that earlier read: “Intriguingly, this difference did not lead to a worse OS ” now reads: “Intriguingly, this difference did not lead to a worse OS as in our study, that included only metastatic UC”

This manuscript is a resubmission of an earlier submission. The following is a list of the peer review reports and author responses from that submission.